# Chitosan Oligosaccharide Fluorinated Derivative Control Root-Knot Nematode (*Meloidogyne incognita*) Disease Based on the Multi-Efficacy Strategy

**DOI:** 10.3390/md18050273

**Published:** 2020-05-22

**Authors:** Zhaoqian Fan, Yukun Qin, Song Liu, Ronge Xing, Huahua Yu, Pengcheng Li

**Affiliations:** 1Key Laboratory of Experimental Marine Biology, Center for Ocean Mega-Science, Institute of Oceanology, Chinese Academy of Sciences, No. 7 Nanhai Road, Qingdao 266071, China; fzq3707@163.com (Z.F.); sliu@qdio.ac.cn (S.L.); xingronge@qdio.ac.cn (R.X.); yuhuahua@qdio.ac.cn (H.Y.); 2Laboratory for Marine Drugs and Bioproducts, Pilot National Laboratory for Marine Science and Technology (Qingdao), No. 1 Wenhai Road, Qingdao 266237, China

**Keywords:** chitosan oligosaccharide (COS), modification, *Meloidogyne incognita*, multi-efficacy, nematicide

## Abstract

Plant root-knot nematode disease is a great agricultural problem and commercially available nematicides have the disadvantages of high toxicity and limited usage; thus, it is urgent to develop new nematicides derived from nature substances. In this study, a novel fluorinated derivative was synthesized by modifying chitosan oligosaccharide (COS) using the strategy of multiple functions. The derivatives were characterized by FTIR, NMR, elemental analysis, and TG/DTG. The activity assays show that the derivatives can effectively kill the second instar larvae of *Meloidogyne incognita* in vitro, among them, chitosan-thiadiazole-trifluorobutene (COSSZFB) perform high eggs hatching inhibitory activity. The derivatives can regulate plant growth (photosynthetic pigment), improve immunity (chitinase and β-1,3-glucanase), and show low cytotoxicity and phytotoxicity. According to the multi-functional activity, the derivatives exhibit a good control effect on plant root-knot nematode disease *in vivo*. The results demonstrate that the COS derivatives (especially fluorinated derivative) perform multiple activities and show the potential to be further evaluated as nematicides.

## 1. Introduction

Root-knot nematode (RKN; *Meloidogyne* spp.) is a pathogen that parasitizes the root of plant and causes the root to expand into a knot. The knots hinder the transmission of water and nutrients, thus affecting the growth of the plant [1]. As a soil borne disease, RKN has a broad host range [2], including a multitude of vegetables and crops, and is difficult to control. In addition, RKN is easily spread by water, machinery, animals, and humans, resulting in huge losses (about 100 billion dollars) to global agricultural production every year [2]. At present, chemical control is the main control measure, but many nematicides are banned because of safety, ecological, and environmental concerns [3]. However, there are many shortcomings in the limited control categories, such as fosthiazate, which can lead to high toxicity to non-target organisms [4] and drug resistance [5,6] and avermectin, which has difficulty to stay structurally unstable [7] and poor mobility in soil. Therefore, it is critical to develop novel types of eco-friendly and highly efficient nematicides.

It is a general trend to develop pesticides by using natural active substances, such as botanical materials [8,9,10], bacteria [11,12], and marine biomolecules [13]. In addition, the outbreak of nematode disease is closely related to soil environment with its complex ecosystem, nematode control needs to take into account ecological, environmental, economic, yield, and other factors. Therefore, the development of multi-functional active substances, which has rarely been considered before, can be used as a research strategy for new green nematicides. These kinds of nematicides not only have nematicidal activity, but can also improve plant resistance and promote its growth. Multiple functions will ensure or enhance the control effect of nematode diseases. Thus, using them will provide an effective control under economic, environmental, and low toxicity conditions, in conformity with a sustainable agriculture. However, there are very few active substances, which can meet the above conditions at the same time. One of the ways to solve this problem is modification (active splicing). As a starting point is to find natural compounds with one or more biological activities and easy to modify. The marine bio-stimulants, such as seaweed polysaccharide and chitosan/chitooligosaccharide, are such substances.

Chitooligosaccharide (COS) is derived from the basic polysaccharide chitin or chitosan, one of the main sources of which is the shells of marine arthropods such as shrimp and crab. COS is low in toxicity, biodegradable, and biocompatible. It can promote plant growth, improve plant immunity and stress resistance, inhibit bacteria/fungi, and is antiviral [14]. Some studies have found that chitin and chitosan can inhibit pathogenic nematodes when applied to the soil [15,16,17]; therefore, COS has the potential to control nematode diseases. However, as far as we are aware, research work on the use of biomolecular COS as a nematicide to control nematode disease have been barely reported in the published literature; low nematicidal activity may be one reason.

In order to apply COS to nematode control based on multi-efficacy strategy, the first step is to break through the bottleneck of weak nematicidal activity. Modification is a reasonable method [18]. The key is to choose workable functional groups. At present, a lot of trifluorobutylene heterocyclic derivatives (such as C1 [19], C2 [20], and C3 [21]) are known to possess high nematicidal activity, and such kind of nematicide (fluensulfone) have been approved for industrial use. The fluorinated nematicides have the advantages of lower toxicity than older nematicides (e.g., fumigants, organophosphates, and carbamates) [22], high safety, and no evidence of enhanced biodegradability [23,24]. Therefore, the fluorinated group, such as trifluorobutylene with heterocycle ring (see Figure 1), is a potential active group.

In this study, following the strategy of multiple effects and the principle of click chemistry, COS is modified by grafting functional groups, such as thiadiazole and trifluorobutylene moieties. As a result, the nematicidal activity can be improved, and the derivatives would possess the immune induce activity of COS moieties and low toxicity. To verify whether the derivatives has multiple functions, the nematicidal activity against J2s and egg hatching inhibitory activity was estimated in vitro, the immune enzyme (chitinase and β-1,3-glucanase) was to evaluate for indicating plant immune change, the photosynthetic pigments tests show the influence on plant growth, and the phytotoxicity and cytotoxicity were investigated for biosafety. Lastly, the greenhouse test tube assay will reflect the actual control effect of multi-functional COS derivatives on root-knot nematode. Furthermore, the derivatives were characterized by FTIR, ^1^H NMR, ^13^C NMR, elemental analysis, and thermogravimetric analyses (TG)/derivative thermogravimetric analyses (DTG). The purpose of this study is to prepare multi-functional derivatives from marine bio-polysaccharide for obtaining novel nematicidal agents with good performance.

## 2. Results

### 2.1. Characterization of FTIR, NMR, Elemental Analysis, and TG/DTG

The COS derivatives were synthesized as Scheme 1, and the structure was characterized by Fourier transform infrared (FTIR), NMR, elemental analysis, and thermogravimetry (TG)/differential thermogravimetry (DTG).

#### 2.1.1. FTIR

As shown in Figure 2, comparing to COS, there is a new wide absorption peak (1542 cm^−1^) assigned to C=S and NH–NH groups belonging to TCDCOS [25]. Moreover, the signal at 945 cm^−1^ due to C=S–S group is weak, illustrating that most C=S–S groups have reacted with hydrazine. Since they have similar main groups, the spectrum of DTCOS is similar to that of TCDCOS. For DTCOS, the difference is that the stretching vibration peak of the C=S group (1545 cm^−1^) is stronger than that of the NH–NH group (1582 cm^−1^), which indicates that new C=S–S groups have been grafted onto NH–NH groups. For COSCDZ, all the absorption peaks of C=S–S, C=S, and NH–NH groups disappeared, and a new peak at 1524 cm^−1^ attributed to C=N moiety appeared. This change indicates that the thiadiazole group has been synthesized. There are new peaks 1245 cm^−1^ (C–F) and 1524 cm^−1^ (C=C) belonging to end-product chitosan-thiadiazole-trifluorobutene (COSSZFB), which show that the trifluorobutylene group has been grafted. In addition, the weak peak of C–S moiety at 837 cm^−1^ indicates that the grafted site is on the –SH of the thiadiazole group.

#### 2.1.2. ^1^H NMR and ^13^C NMR

The functional groups of TCDCOS, DTCOS, and COSCDZ have no hydrogen atoms; thus, the ^1^H NMR spectra were similar to that of COS (Figure 3A). By contrast, the spectrum of COSCZFB changed significantly. For example, new peaks of S–CH_2_ and –CH_2_ were observed at δ 4.3 ppm and δ 3.2 ppm, respectively. In addition, the impurity peaks are ascribed to ethanol (δ 3.5 ppm, δ 1.0 ppm), dimethyl sulfoxide (δ 2.5 ppm), and deuterium acetic acid (δ 1.8 ppm). 

For the analysis of carbon-containing and hydrogen-free groups, ^13^C NMR is more practical than ^1^H NMR. As shown in Figure 3B, the C=O moiety of COS was at δ 180.1 ppm; however, the peaks of other derivatives were weakened because of the removal of impurities. For DTCOS and TCDCOS, new signals at δ 177.8 ppm are ascribed to C=S moieties. In the spectrum of COSCDZ, this signal disappeared, and a new peak at δ 174.5 ppm assigned to C=N moiety is observed. This shows that the thiadiazole group has been obtained. For COSCZFB, the shifts of new moieties appear at δ 174.5 ppm (C=C), δ 84.3 ppm (S–CH_2_), and δ 62.2 ppm (–CH_2_–). The results demonstrate that the target compounds have been synthesized successfully.

#### 2.1.3. Elemental Analysis

The deacetylation degree (DD) and substitution degree (DS) of the derivatives (except COSCDZ) were calculated according to the integrations of peaks in ^1^H NMR spectra and the C/N value obtained by elemental analysis, respectively. The following equations were used:(1)$$DD\left(\%\right)=1-\frac{I_{CH3}}{3\times I_{H2}}\times 10$$
(2)$$\frac{12\times \left[6+2\times \left(1-DD_{x}\right)+\Sigma \left(n\times DS_{x}\right)\right]}{14\times \left(1+2\times DS_{1}\right)}={\left(\frac{C}{N}\right)}_{x}$$
where I_CH3_ and I_H2_ are the integrations of methyl and H2, respectively, *x* represents the product in each step, and *n* is the carbon number of new groups, which are 1, 1, and 4, respectively.

The C/N value can reflect the changes occurring to the groups. For TCDCOS, DTCOS, and COSCZFB, the C/N values indicate the increase or decrease law of carbon and nitrogen atoms due to the deacetylation and substitution. However, for COSCDZ, the C/N value, which should be similar to that of DTCOS, increased significantly, showing that the thiourea groups were lost under acid condition. Therefore, the DS of the COSCDZ thiadiazole group cannot be calculated using the equations above. For COSCZFB, the DS of trifluorobutenyl can also be calculated by the integration of S-CH_2_ (3.86) and H2 (5.85), and the result was 33.0% which was near the value reported in Table 1, illustrating the accuracy of the DS value.

#### 2.1.4. TG/DTG

As shown in Figure 4, the TG/DTG curves of derivatives are similar to that of COS and contain three–four decomposition peaks. The mechanisms for two peaks are the same, one is at 40–60 °C due to the loss of water and volatile substances, the other is at 260–280 °C ascribed to the depolymerization and degradation of sugar ring molecules. In addition to molecular chain ablation, the causes for weight loss in the middle stage also include dissociation, decomposition, and ablation of grafted groups, which lead to different decomposition peak temperature (*Td*). For example, the weight loss (1.7%) of TCDCOS at 156 °C is due to the dissociation of the free hydrazine group, and the effect at 227 °C is ascribed to decomposition of the thioacyl group. For DTCOS, 8.6% of the substances lost are hydrazine and sulfhydryl groups. The reason for the loss at 238 °C is the same as for TCDCOS (227 °C). The thiazole group of COSCDZ is a five-membered ring, which is a stable structure, since the *Td* (198 °C) is high. Meanwhile, there are no other substituent groups and no other *Td*. The trifluorobutylene group provides more hydrogen bonds which make its structure more stable; hence, two main *Td* of COSCZFB are higher than other derivatives. The weigh losses of derivatives at the second stage are between 11% and 20%, which are consistent with the content of grafted groups. 

### 2.2. Egg Hatch Inhibitory Activity

In the process of controlling nematode diseases by nematicides, inhibiting egg hatching is an important way. As shown in Figure 5, the inhibitory activity of COSCZFB on *M. incognita* eggs hatching increased with increasing concentration and incubation time. Especially at 14 days, the HI value at 1 mg/mL exceeded 90%, which was significantly higher than the maximum value of COS (60%). At other stages, the activity was also higher than that of COS. Therefore, the thiadiazole-S-trifluorobutene group improved the egg hatch inhibitory activity of COS. However, the activity changes of TCDCOS, DTCOS, and COSCDZ were complex, it is difficult to say that the relative groups (thiourea, dithioformyl, and thiadiazole) have a positive influence on COS.

### 2.3. Nematicidal Activity against J2s

The second-rate larval stage is an important stage for nematode control. As shown in Table 2, the nematicidal activity of fluorine-containing derivatives on the J2s increases with concentration and time. The LC_50_ of each derivative is less than 1 mg/mL, indicating excellent nematicidal activity compared with COS. Among the compounds, TCDCOS, COSCDZ, and COSCCZFB have higher activities. At 0.25 mg/mL, the mortalities of derivatives in 24 h are over 60%. 

For TCDCOS, the thiourea group plays an important role in the nematicidal activity; thus, the mortality decreases after grafting dithiocarbamate groups to synthesized DTCOS. As a result, the LC_50_/48 h of TCDCOS increases from 0.10 mg/mL to 0.68 mg/mL. The thiadiazole ring is one of the main structures of several insecticides with strong biological activity. In this study, the thiadiazole ring showed stronger nematicidal activity than the thiourea group, according to the lethal rate of COSCDZ at 0.25 mg/mL, which is higher than that of TCDCOS. Additionally, COSCZFB has the highest nematicidal activity among all the derivatives, and its nematode mortality even reaches 100% at a concentration of 0.25 mg/mL. At the same concentration, the activity is near to that of the positive control fluenesulfone. Trifluorobutenyl compounds have certain nematicidal activity [26,27,28]. When trifluorobutylene and thiadiazole coexist, the activity of chitosan oligosaccharides is further improved.

In summary, the LC_50_/72 h of each derivative is much lower than that of COS; thus, the fluorine-containing derivatives and its intermediate have better nematicidal activity than COS, that is, thiourea, thiadiazole, and trifluorobutylene groups can improve the nematicidal activity of COS. Therefore, these derivatives have great potential for industrial application.

### 2.4. Actual Control RKN Disease Ability

As shown in Figure 6A,B, the disease indices (Dis) of cucumber seedlings decrease significantly after the application of the derivatives, and the values also decrease with increasing concentration, indicating that all the samples have a certain control effect on the *M. incognita* disease. Among the derivatives, COSSZFB has the best activity at 1 mg/mL, which can rival fluensulfone and fosthiazate. The result is consistent with the in vitro experiment. It can be seen in Figure 6C that the number of root knots in the COSSZFB treatment group is significantly less than that in the control group and the COS treatment group, and the difference between the COSSZFB treatment group and positive control group is not significant. The activities of other samples are weaker than that of the positive control, but it is slightly stronger or equivalent than that of the COS. Application 7 days before the inoculation of nematode can make the effect more obvious, and the DI of DTCOS can even reach that of the positive control. It is also found that it takes time for these agents to exert their optimum activities. This is different from COSSZFB, which has a shorter application time but a stronger control effect. The action mechanism of the double derivatives is different. It is speculated that the former (TCDCOS, DTCOS, and COSDZ) need degradation to release the active groups while the latter (COSSZFB) can play a direct role in contract killing, which needs further confirmation. To summarize, the introduction of active groups improved the control effect of COS on RKN. Figure 6D shows that cucumber seedlings grew well in the sample treatment group, while in the positive control group, yellow spots appeared in the leaves, and the seedlings grew weakly, demonstrating the lower phytotoxicity of derivatives.

### 2.5. Effect on Photosynthetic Pigment Content

As shown in Table 3, fluorinated derivatives and intermediates can promote the synthesis of photosynthetic pigments in tomatoes. The amount of photosynthetic pigment is highest in the COSCDZFB group at 0.8 mg/mL, followed by COS, COSCDZ, and TCDCOS groups, which are similar or higher than those of blank control. This indicates that the trifluorobutylene group promoted the plant growth regulation ability of COS, while the thiourea group had little effect. In other words, the introduction of the thiourea group and trifluorobutylene group did not weaken the plant growth regulation ability of COS, nor enhance the toxicity to plant growth. 

### 2.6. Effect on Immune Enzyme

Chitin is one important part of plant pathogenic fungi cytoderm, nematode body wall, and eggshell. Several studies show that chitinases can affect cuticle and eggshell in the infection process of nematophagous fungi [12,29,30]. The application of chitin or chitosan in the soil can increase the bacteria chitinase production, which may be one reason for inhibiting the RKN population [17]. Hence, the chitinase, including excision enzyme and incision enzyme, is an important immunity enzyme [31]. Therefore, the effects of COS derivatives on plant chitinase activities were assayed in this study. As shown in Figure 7A, the chitinase activity of each derivative is lower than that of COS, indicating that the graft of nematicidal groups, affects the action of the COS backbone. Fortunately, its influence is not very important. The activities of all compounds are significantly higher than that of the blank control. This illustrates that the derivatives can regulate the chitinase activity of the plant effectively. In addition, the highest activity is not always obtained at the highest concentration of derivatives; thus, the concentration is not a key parameter for improving chitinase activity.

β-1,3-glucanase is another important enzyme that is related to plant defense against soil pathogens including nematode [2,32]. As shown in Figure 7B, it can be seen that the β-1,3-glucanase activities of all the derivatives are better than that of the control group. Comparing to the COS group, the results show that thiourea, thiadiazole, and trifluorobutylene moieties are beneficial to the induction of β-1,3-glucanase. In addition, the maximum effective concentrations of TCDCOS, COSCDZ, COSCZFB, and COSFB are 0.2 mg/mL, 0.8 mg/mL, 0.2 mg/mL, and 0.2 mg/mL, respectively. Therefore, as well as chitinase, the enzyme activity is not positively correlated with the concentration. The reason may be that the COS molecule is an effective plant immunity agent [31,33], and a large amount might not be needed to induce a variety of disease resistance reactions. However, please note that the signal transduction of β-1,3-glucanase is still unclear.

### 2.7. Cytotoxicity

The dendritic sarcoma cells (DCS) cell survival can reflect the cytotoxicity of a chemical compound. As shown in Figure 8, the cell survival rates of TCDCOS, DTCOS, and COSCDZ are more than 100%, representing good biocompatibility. For COSSZFB, the value is more than 90%, which can be considered as having low toxicity on DCS cells. However, it also found that the trifluorobutylene group affects the biocompatibility of COS. Nevertheless, all derivatives show lower cytotoxicity to the mammal cell than fluensulfone, which exhibits a survival value lower than 90% at 0.005 mg/mL. Thus, one might suggest that the COS backbone is beneficial to reduce the cytotoxicity of the active groups. However, more comprehensive acute and chronic toxicity experiments are needed to evaluate the biological safety of the derivatives to mammals.

### 2.8. Phytotoxicity

As shown in Table 4, the percentage germinations of derivatives at different concentrations are similar to that of the blank control, where the results show that the compounds have no adverse effect on seed germination. For root elongation, the influences of derivatives are positive, as can be deduced through an RGI value > 0.8. Therefore, it can be concluded that the derivatives with a GI > 1 had no phytotoxicity on seedlings, confirming the findings above. 

## 3. Discussion

Nematode control is a complex ecological process, involving pathogenic nematodes, hosts (crops), natural enemies, cooperative bacteria or fungi, non-target organisms, soil, and so on. However, existing nematicides easily cause side effects, such as disease resistance, plant or non-target organism toxicity, and even environmental problems. The main purpose of nematode control is to increase crops yield, reduce resource waste, and decrease poisoning. Therefore, the design of multi-functional agents is a good strategy for the development of novel pesticides. The marine polysaccharide with plant bio stimulatory activity could be the source material, such as seaweed polysaccharide and chitin. This study shows that chitin, chitosan, and COS can be considered good nematode control candidates. As low toxic pesticides, chitosan and COS can be manufactured as sustainable pesticides encapsulating materials [34,35], or modified by grafting functional groups [36]. The methods can be the route to deal with the toxicity of nematicide [7,37]. In this study, the latter route, which we believe more convenient and easier, was chosen.

This study demonstrates that the COS derivatives have effective nematicidal activities against *M. incognita*, and COSSZFB also exhibit high egg hatch inhibitory activity. Furthermore, the derivatives possessed plant growth and immune regulation effects due to the COS backbone, which can improve the resistance against RKN and reduce the damage caused by nematode disease. The derivatives (especially COSSZFB) displayed a remarked control effect on cucumber *M. incognita* disease. In addition, the compounds performed low cytotoxicity and phytotoxicity, pointing out that the results of the current study are promising. 

To summarize, the multi-efficacy was achieved by grafting exogenous groups to COS, to improve the nematicidal activity of COS against RKN. According to this study, there is no doubt that these functional groups (thiadiazole, trifluorobutylene moieties) play a major role in the nematode control process. For the COS backbone, the effect of inhibiting egg or J2 is unclear. The way of chitin and chitosan to control nematode in soil may be by ameliorating the micro-environment to promote the growth of nematophagous bacteria and fungi (such as chitinase producing organisms) [38,39], and the release of ammonia during their decomposition [40,41]. In addition, for growth or immune regulation, the COS backbone plays a center role. However, it cannot be confirmed here whether there is a synergistic effect between the COS backbone and the functional groups. While more research work is needed to clarify further the mechanism of COS derivatives to control nematodes, this study demonstrated that the COS derivatives are effective nematicide, especially fluorinated derivative COSSZFB. It shows that the marine biomacromolecules with bio-stimulation activity on plants can be used to develop nematicides. 

## 4. Materials and Methods 

### 4.1. Materials

COS with a degree of deacetylation (DD) of 13% and average molecular weight (MW) of 1500 Da was purchased from Qingdao Yunzhou Biochemical Co., Ltd. Deuterium oxide was purchased from J&K Scientific Ltd. (Beijing, China). 4-Bromo-1,1,2-trifluoro-1-butene (BTF) was purchased from Shanghai Yuanye Biotechnology Co., Ltd. (Shanghai, China). Other chemical reagents, of analytical grade, were purchased from Sinopharm Chemical Reagent Co., Ltd. (Shanghai, China). The Roswell Park Memorial Institute (RPMI) 1640 medium and penicillin-streptomycin solution were obtained from Gibco BRL (Thermo Fisher Scientific (China) Co., Ltd., Shanghai, China). Fetal bovine serum (FBS) was supplied by HyClone (ThermoFisher, Logan, USA). Tomato plants were grown in a phytotron, and the plant samples were harvested after growing from seed to the six-leaf stage in a sterile medium [42]. Mouse dendritic sarcoma cells (DCS) were purchased from American Type Culture Collection (Manassas, VA, USA).

### 4.2. Characterization Methods 

FTIR spectra ranging from 4000 cm^−1^ to 400 cm^−1^ were obtained using a Thermo Scientific Nicolet iS10 spectrometer (Thermo Fisher Scientific (China) Co., Ltd., Shanghai, China). ^1^H NMR and ^13^C NMR were recorded by a JEOL JNM-ECP600 spectrometer (JEOL Ltd., Tokyo, Japan) using D_2_O and CD_3_COOD as solvents. Elemental analysis (C, N, and S) was performed using a Vario EL-III elemental analyzer (Elementar Analysensysteme GmbH, Hanau, Germany). The percentages of C, N, and S were used to calculate the degree of substitution (DS) of the COS derivatives. A METTLER TGA-DSC 1 SF/1382 (Mettler-Toledo international trade (Shanghai) Co., Ltd., Shanghai, China) was used to record TG and DTG curves of the polymers from 25 °C to 500 °C. 

### 4.3. Synthesis of COS Thiadiazole Trifluorobutene Derivative 

As shown in Scheme 1, the derivative TCDCOS was obtained according to the method of Qin, with a 59.6% yield [25]. The post-treatment step was to produce a solid by absolute ethanol precipitation and centrifugation. The solid was dissolved in DMSO, centrifugated at rt, 8000 RPM for 5 min (CT1812RT, Techcomp (China) Ltd.), Beijing, China) to remove salts, and precipitated with ethanol again. After repeating the above step, the precipitate was collected, washed with ethanol, and dried at 60 °C using an oven (XMTD-8222, Shanghai Jinghong Co., Ltd., Shanghai, China). Finally, the chitosan oligosaccharide thiocarbonyl hydrazide (TCDCOS) was obtained. 

TCDCOS (10.0 g) and potassium hydroxide (3.6 g) were added to absolute ethanol (25 mL). Then, 12 mL of carbon disulfide/ethanol mixture (v:v = 1:5) was dripped slowly under stirring at room temperature for 10 h. After filtration, the filter cake was dissolved in DMSO and the residue was removed by centrifugation at rt, 8000 RPM for 5 min. The precipitate was obtained again by alcohol precipitation. The post-treatment processes to produce Chitosan oligosaccharide formyl hydrazide dithioformate (potassium) derivative (DTCOS) were the same as that of TCDCOS with a yield of 65.7%. 

To produce Chitosan oligosaccharide-thiadiazole derivative (COSCDZ), DTCOS (5 g) was dissolved in 20 mL of distilled water, and the solution was adjusted to pH = 2–3 with 3 mol/L of sulfuric acid. After stirring for 30 min, 50 mL of ethanol was added to obtain a precipitate. The next step was the same as that for the treatment of TCDCOS, yielding 49.8%. 

Next, COSCDZ (2 g) and 4-bromo-1,1,2-trifluoro-1-butene (BTFB) (1.6 mL) were added to 20 mL of acetonitrile. Triethylamine (1.8 mL) was added and stirred for 0.5 h. Then, the mixture was heated to 60 °C and reacted for 12 h. After filtration, the filter cake was washed twice with absolute ethanol and dissolved into water before freeze-drying. Chitosan-thiadiazole-trifluorobutene derivative (COSSZFB) was obtained with a yield of yield of 76.6% (total yield from COS was 14.9%).

### 4.4. Egg Hatching Assay In Vitro

The nematicidal activities of the derivatives were estimated against the nematodes (*M. incognita*) isolated from the diseased cucumber plants in greenhouses in Qingdao, China. The eggs collection and the second instar larvae (J2s) incubation were according to the method described by Bogner et al. [43]. The egg hatch inhibitory activities of derivatives were estimated by the dipping method. Briefly, egg suspension (20 μL, 100 eggs) and compound solution (200 μL) were added into each well of a 48-well plate. Then, moderate sterile distilled water (280 μL) was added to achieve final concentrations of 6, 3, 2, 1, 0.5, and 0.25 mg/mL. The plates were covered with films and placed at 28 °C for incubation. The number of juveniles was countered at 3, 7, and 14 d. In the tests conducted with four replicates, distilled water and fluensulfone were used as blank control and positive control, respectively. 

The rate of egg hatching (HR) was calculated using:(3)HR=[juveniles/(eggs+juveniles)]×100

The egg hatching inhibitory index (HI) was calculated with the corrected HR following the formula:(4)HI (%)=[(C−T)/C]×100
where *C* and *T* represent the HR of blank control and treatment, respectively.

### 4.5. Nematicidal Assay In Vitro 

In this study, the nematicidal activities of derivatives were evaluated by the dipping method [44]. For this, nematode suspension containing 100 J2s (20 μL) and sample stock solution (10 mg/mL) were added into a well of 24-well plate; then, distilled water was added to give a final volume 500 μL. For each sample including positive control (fluthiazone), there were three concentration gradients (1, 0.25, and 0.063 mg/mL) and four repeats. After culturing for 24, 48, and 72 h under dark conditions at 28 °C, the mortality (%) rate was counted. If the nematode body was motionless or stiff, it was considered dead.

The nematicidal activity of the derivative was expressed by the corrected mortality, which was calculated by the following formula:(5)$$\mathrm{Corrected}\;\mathrm{mortality}\;(\%)=\frac{mortality\;in\;treatment-mortality\;in\;blank\;control}{100-mortaliy\;in\;blank\;control}\times 100$$

The data were analyzed by ANOVA according to Duncan’s method. It was considered statistically significant when *p* < 0.05. The LC_50_ values were calculated using IBM SPSS Statistics 21.0 software based on probit regression. 

### 4.6. Greenhouse Test Tube Assay In Vivo

For preliminary estimation of the actual activities of derivatives to control nematode disease, the method of test tuber assay in vivo was adopted according to the method of Wahla [45] and Andrés [46], with appropriate modification. Briefly, the cucumber seed after germination was planted into a test tube (15 mm × 100 mm) with 1/3 coarse sand and cultivated in the greenhouse for one week. Then, 0.5 mL of sample solution (1 and 0.5 mg/mL) or positive solution (fosthiazate and fluensulfone in 0.05 mg/mL) was added to the test tube. Then, 20 μL of nematode solution (40–50 worms) were inoculated on the first, third, and seventh day after sample application, respectively. Twenty days later, the numbers of root knots were counted. Water was the blank control and each treatment was replicated three times. The disease classification (DC) standard was described as follows: Level 0 represents no visible root knot; Level 1, 2, 3, and 4 represent the root system with root knot forming lumps of 1%–25%, 26%–50%, 51%–75%, and 76%–100%, respectively. 

The effect of prevention of derivative on cucumber nematode disease was estimated by disease index (DI) and control effect rate (CE), which were calculated by the following formula:(6)$$\mathrm{DI}=\frac{\sum^{\;}\;\left(\mathrm{DC}\;value\times the\;number\;of\;disease\;plants\;on\;relative\;\mathrm{DC}\right)}{total\;number\;of\;investigated\;plants\times maximal\;\mathrm{DC}\;value}\times 100$$
(7)$$\mathrm{CE}\;\left(\%\right)=\frac{\mathrm{DI}\;of\;blank\;control-\mathrm{DI}\;of\;treatment}{\mathrm{DI}\;of\;blank\;control}\times 100$$

### 4.7. Plant-Regulation Assays

In order to evaluate the effect of COS derivative on plant physiology, the contents of chlorophyll a, b and carotenoid in tomato seedlings were determined by the method of Zong [47]. Tomato seedlings with the same growth rate of six leaves were divided into three groups, four seedlings in each group were sprayed with 50 mL of the compound solution (0.2, 0.4, and 0.8 mg/mL) or water (blank control). After spraying with the three times in 6 days, 0.2 g of fresh leaf tissue was extracted and ground in a mortar with a 15 mL of 95% ethanol, quartz sand and calcium carbonate. The slurry was diluted to 25 mL for determination of chlorophyll a (Chl a), chlorophyll b (Chl b), total chlorophyll (a + b), and carotenoid (Car). The absorbances at 665, 649, and 470 nm were measured by a spectrophotometer (TU1810, Beijing Puxi Ltd., China). The contents of pigments were calculated according to Equations (8)–(10):Chl a = 13.95 *A*_665_ − 6.88 *A*_649_(8)
Chl b = 24.96 *A*_649_ − 7.32 *A*_665_(9)
Car = (1000 *A*_470_ − 2.05 Chl a − 114.8 Chl b)/245(10)

### 4.8. Chitinase and β-1,3-glucanase Assay 

Defense enzyme (chitinase and β-1,3-glucanase) activities were determined for assessing the effects of the derivatives on plant immunity [42,48]. The tomato plants were treated as in plant-regulating assays. The fresh leaves were frozen with liquid nitrogen and ground into powder. In total, 0.1 g of powder and 2 mL of acetate buffer (0.05 mol/L, pH 5.0) were added into an Eppendorf tube and centrifugated at 4 °C, 12,000 RPM for 15 min. The supernatant is the rough enzyme extract.

For chitinase, 0.4 mL of enzyme extract, 0.4 mL of acetate buffer and 0.4 mL of colloidal chitin solution (1%) were made into a mixture solution and held in a water bath at 37 °C for 2.5 h. After centrifugation at 5000 RPM for 10 min, 0.4 mL of supernatant was mixed with 40 μL of snail enzyme solution (1%), followed by heating in a water bath at 37 °C for 0.5 h. Then, a saturated borax solution (0.2 mL) was added into the mixture and boiled for 7 min, followed by cooling immediately. Next, glacial acetic acid (2 mL) and DMAB (1 mL, 1%) were added and the mixture was kept in a water bath (37 °C) for 15 min. Finally, the absorbance of the solution at a wavelength of 585 nm was measured. According to the standard curve, the absorbance was converted to the amount of N-acetylglucosamine produced, i.e., the total chitinase activity. One enzyme activity unit (U) can be defined as the amount of enzyme needed to decompose colloidal chitin to produce 1 μg N-acetylglucosamine in unit weight (g) of fresh plant tissue and unit time (hour).

The activity of β-1,3-glucanase was determined by the dinitrosalicylicacid (DNS) method. Briefly, 0.4 mL crude enzyme extract was mixed with 0.4 mL of laminarin solution (1 mg/mL). The mixture was kept at 37 °C for 30 min, then held in a boiling water bath for 10 min to terminate the reaction, followed by cooling to room temperature. The mixture was added with 0.8 mL of DNS reagent and kept in a boiling water bath for 10 min. After cooling with cold water, the mixture was mixed with 2.4 mL of distilled water, and the absorbance was measured at 550 nm. According to the standard curve, the absorbance that represented the β-1,3-glucanase activity was converted into the amount of glucose produced. One unit of enzyme activity (U) was defined as the amount of enzyme needed to produce 1 mg of glucose per minute in unit weight (g) of fresh plant tissue.

### 4.9. Cell Toxicity

Thiazolyl Blue (MTT) method was used to detect the cytotoxic activity of COS derivative against mouse DCS cells. Briefly, DCS cells (3 × 10^4^ cells/well) were seeded on a 96-well plate. After overnight incubation in an incubator at 37 °C, fresh media containing 0.2 mg/mL of compound solution was added. After 24 h, the supernatant was carefully removed from each well. Next, DCS cells were incubated in 10 μL MTT reserve solution (5 mg/mL PBS) and 90 μL of FBS free medium. After 4 h at 37 °C, 100 μL of MTT stop solution (10% sodium dodecyl sulfate and 0.01M hydrochloric acid) was added to dissolve the formamide crystals. Finally, the absorbance at 490 nm was measured with a spectrophotometer (Tecan, Mannedorf, Switzerland). The cell survival rate is the ratio of absorbances of the sample and the untreated group.

### 4.10. Phytotoxicity 

For estimating phytotoxicities of the derivatives, the effects on seed germination and root elongation were assayed by the method of Magdaleno et al. [49]. Briefly, 10 uniform and full cucumber seeds were selected and placed in a 90 mm diameter Petri dish lined with a filter paper. Then, 4 mL of sample solution (0.5, 1.0 and 2.0 mg/mL) was added. The seeds were incubated at 25.0 ± 2.0 °C in darkness. After 48 h, the seed germination and radicle elongation were counted. Each treatment was replicated three times and distilled water was used as a blank control. The germination index (GI) and root elongation index (RGI) were calculated using the following equations:RGI = RL_S_/RL_C_(11)
GI (%) = (RL_S_ × GS_S_)/(RL_C_ × GS_C_)(12) where the subscripts S and C represent the results of the sample treatment group and blank control group, respectively, and RL and GS express the radicle lengths of the seeds and the numbers of germinated seeds, respectively. RGI values between 0.8 and 1.2 indicate that there is no effect of the derivatives on the root elongation. When GI exceeds 1, the derivative is considered to be not phytotoxic.

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
