# Peer review of "Chitosan Oligosaccharide Fluorinated Derivative Control Root-Knot Nematode (*Meloidogyne incognita*) Disease Based on the Multi-Efficacy Strategy"

_marinedrugs, 2020, doi:10.3390/md18050273_

Round 1
Reviewer 1 Report
The manuscript contains interesting and important data. However, I have some remarks, listed below. The manuscript is unclear in some places. The introduction, in my opinion, does not describe precisely the idea of the work, does not explain why the particular experiments were done. Please develop a discussion and justify your research plan.
Line 56. What “ inhibit bactericidal” means? Unclear
Line 67. “long shelf-life (no evidence of enhanced biodegradability)” – the long half-life is often a disadvantage (the most drastic example – DDT). Therefore, the statement is not clear. I propose that the authors may specify the time range they are talking about.
Line 72 The purpose of this study is to obtain novel nematicidal agents with good performance from marine bio-polysaccharide”. – the authors also tried to estimate the activity of the obtained agents.
Lines 78-80 Delete the editorial remark “This section may be divided by subheadings. It should provide a concise and precise description of the experimental results, their interpretation as well as the experimental conclusions that can be drawn.”
Line 81: “Characterization” – of? Please extend the title.
Figure 5 seems to be a little bit confusing. For example, if you read the value for COSCZFB, you will see almost 90% inhibition on the 1st day, less than 70% on the 7th, and again about 90% on the 14th day. Is it a total hatching success or the data refer to a particular day?
Line 168 time. The LC50 of each derivative is less than 1 mg/mL, indicating excellent nematicidal activity - this statement must be supported by comparison to commercial nematicides. The authors compare the activity to COS, which is obviously ok, but they should always also refer it to the positive control, too.
Thiourea is toxic (endocrine disruptor). Also, the authors write that “Thiadiazole ring is one of the main structures of several insecticides with strong biological activity”. Can we avoid soil pollution and make the activity-specific?
Figure 6D shows that cucumber seedlings grew well in the sample treatment group, while in the positive control group, yellow spots appeared in the leaves, and the seedlings grew weakly, demonstrating the lower phytotoxicity of derivatives. – please indicate/mark these changes on the figure.
Line 255. “For COSSZFB, the value is more than 90%, which can be considered as having low toxicity on mammal cells. However, it also found that the trifluorobutylene group affects the biocompatibility of COS. Nevertheless, all derivatives show lower cytotoxicity to a mammal than fluensulfone,” – what data on mammals do the authors have on mind? This expression is not precise. Do the authors think about mammalian cells they tested or mammal cells in total? The authors just checked the toxicity against one cell type.
Line 379 “After 1, 3 and 7 d, 20 μL of nematode solution” – after a solution? Unclear
Line 391 chlorophyll a, b and carotenoid in tomato seedlings were determined by the method of zong [47] – “Zong”
Line 392 Tomato seedlings with the same growth rate of 6 leaves were divided into 3 groups, each group was sprayed with the compound solution (0.2, 0.4 and 0.8mg/mL) or water (blank control). – What was the volume sprayed?
Line 403 2.6 and 4.8. Immune assay – be precise: chitinase and β-1,3-glucanase assays.
Why 4.7 and 4.8 are performed on tomato, not cucumbers? Besides, the results present results on cucumbers.
Line 346. Egg hatching assay in vitro – how much eggs were taken to the tests?
Line 442 “…were selected and placed in a 90 mm diameter petri dish” – Petri dish
To sum up, I suggest a major revision of the text.
Author Response
Please see the attachment, thanks!

Reviewer 2 Report
General comments:
In my opinion, these results are really interesting however some point need more attention:
- The main mistake in the whole manuscript (also in the article title) is discussing new materials as chitosan oligosaccharide derivatives. As given in the manuscript (line 305, p. 13) the degree of deacetylation is substantially lower than 50% suggests that it was rather chitin oligosaccharide (CTOS).
- It is also worth noting that CTOS with a low degree of polymerization (DP) ranging from 2 to 6 can be dissolved in neutral water, but CTOS with DP > 6 is not, which limits its application. The molecular weight of CTOS used in this study suggests that DP > 6. Please discuss how changes the solubility of CTOS after modification. In different studies “compound solution”/”stack solution” was used – in what kind of solvent? Discuss changes in solubility.
- It is well known that CTOS production is still challenging with a small number of commercial products using chemical methods. Enzymatic production of CTOS is not very successful. In this context how it is possible to produce enough CTOS for its modification? What is the cost of 10g CTOS production?
Particular comments:
- Calculate the total yield of COSSZFB synthesis from CTOS
- If amino groups of CTOS are incorporated into reaction thus why the degree of deacetylation changes substantially? Are the acetyloamino groups involved in the modification? Explain in detail. Even if amino groups react and thiourea groups are lost under acidic conditions, this parameter shouldn’t be affected. The calculation of DD is unclear.
- Correct name in Scheme 1 (DCTCDCOS into DTCOS) as in 4.3
- In Scheme 1 COS structure should be corrected as if DD<13% there are more acetyloglucose units than amino-ones.
- Line 353: correct “coved” into “covered”
- Line 391: correct “zong” into “Zong”
Author Response
Please see the attachment, thanks!

Round 2
Reviewer 1 Report
The authors responded to all my remarks.